# Coordinated Exploration via Intrinsic Rewards for Multi-Agent Reinforcement Learning

## Abstract

Solving tasks with sparse rewards is one of the most important challenges in reinforcement learning. In the single-agent setting, this challenge has been addressed by introducing intrinsic rewards that motivate agents to explore unseen regions of their state spaces. Applying these techniques naively to the multi-agent setting results in agents exploring independently, without any coordination among themselves. We argue that learning in cooperative multi-agent settings can be accelerated and improved if agents coordinate with respect to what they have explored. In this paper we propose an approach for learning how to dynamically select between different types of intrinsic rewards which consider not just what an individual agent has explored, but all agents, such that the agents can coordinate their exploration and maximize extrinsic returns. Concretely, we formulate the approach as a hierarchical policy where a high-level controller selects among sets of policies trained on different types of intrinsic rewards and the low-level controllers learn the action policies of all agents under these specific rewards. We demonstrate the effectiveness of the proposed approach in a multi-agent gridworld domain with sparse rewards, and then show that our method scales up to more complex settings by evaluating on the VizDoom (Kempka et al., 2016) platform.

## 1 Introduction

Recent work in deep reinforcement learning effectively tackles challenging problems including the board game Go (Silver et al., 2016), Atari video games (Mnih et al., 2015), and simulated robotic continuous control (Lillicrap et al., 2016); however, these successful approaches often rely on frequent feedback indicating whether the learning agent is performing well, otherwise known as dense rewards. In many tasks, dense rewards can be difficult to specify without inducing locally optimal but globally sub-optimal behavior. As such, it is frequently desirable to specify only a sparse reward that simply signals whether an agent has attained success or failure on a given task. Despite their desirability, sparse rewards introduce their own set of challenges.

When rewards are sparse, determining which of an agent's actions led to a reward becomes more difficult, a phenomenon known in reinforcement learning as the credit-assignment problem. Furthermore, if rewards cannot be obtained by random actions, an agent will never receive a signal through which it can begin learning. As such, researchers have devised methods which attempt to provide agents with additional reward signals, known as intrinsic rewards, through which they can learn meaningful behavior (Oudeyer & Kaplan, 2009). A large subset of these works focus on learning intrinsic rewards that encourage exploration of the state space (Pathak et al., 2017; Houthooft et al., 2016; Burda et al., 2019; Ostrovski et al., 2017; Tang et al., 2017).

Exploring the state space provides a useful inductive bias for many sparse reward problems where the challenge lies in "finding" rewards that may only be obtained in parts of the state space that are hard to reach by random exploration. These exploration-focused approaches frequently formulate their intrinsic rewards to measure the "novelty" of a state, such that agents are rewarded for taking actions that lead to novel states. Our work approaches the question of how to apply novelty-based intrinsic motivation in the cooperative multi-agent setting.

Directly applying novelty-based intrinsic motivation to the multi-agent setting results in agents each exploring their shared state space independently from one another. In many cases, independent exploration may not be the most efficient method. For example, consider a task where multiple agents are placed in a maze and their goal is to collectively reach all of the landmarks that are spread out through the maze. It would be inefficient for the agents to explore the same areas redundantly. Instead, it would be much more sensible for agents to "divide-and-conquer," or avoid redundant exploration. Thus, an ideal intrinsic reward for this task would encourage such behavior; however, the same behavior would not be ideal for other tasks. For example, take the same maze but change the task such that all agents need to reach the *same* landmark. Divide-and-conquer would no longer be an optimal exploration strategy since agents only need to find one landmark and they all need to reach the same one. Cooperative multi-agent reinforcement learning can benefit from sharing information about exploration across agents; however, the question of what to do with that shared information depends on the task at hand.

In order to improve exploration in cooperative multi-agent reinforcement learning, we must first identify what kinds inductive biases can potentially be useful for multi-agent tasks and then devise intrinsic reward functions that incorporate those biases. Then, we must find a way to allow our agents to adapt their exploration to the given task, rather than committing to one type of intrinsic reward function. In this work, we first introduce a candidate set of intrinsic rewards for multi-agent exploration which hold differing properties with regards to how they explore the state space. Subsequently, we present a hierarchical method for simultaneously learning policies trained on different intrinsic rewards and selecting the policies which maximize extrinsic returns. Importantly, all policies are trained using a shared replay buffer, drastically improving the sample efficiency and effectiveness of learning in cooperative multi-agent tasks with sparse rewards.

## 2 RELATED WORK

**Single-Agent Exploration** In order to solve sparse reward problems, researchers have long worked on improving exploration in reinforcement learning. To achieve these means, prior works commonly propose reward bonuses that encourage agents to reach novel states. In tabular domains, reward bonuses based on the inverse state-action count have been shown to be effective in speeding up learning (Strehl & Littman, 2008). In order to scale count-based approaches to large state spaces, many recent works have focused on devising pseudo state counts to use as reward bonuses (Bellemare et al., 2016; Ostrovski et al., 2017; Tang et al., 2017). Alternatively, some work has focused on defining intrinsic rewards for exploration based on inspiration from psychology (Oudeyer & Kaplan, 2009; Schmidhuber, 2010). These works use various measures of novelty as intrinsic rewards including: transition dynamics prediction error (Pathak et al., 2017), information gain with respect to a learned dynamics model (Houthooft et al., 2016), and random state embedding network distillation error (Burda et al., 2019).

**Multi-Agent Reinforcement Learning (MARL)** Multi-agent reinforcement learning introduces several unique challenges that recent work has attempted to address. These challenges include: multi-agent credit assignment in cooperative tasks with shared rewards (Sunehag et al., 2018; Rashid et al., 2018; Foerster et al., 2018), non-stationarity of the environment in the presence of other learning agents (Lowe et al., 2017; Foerster et al., 2018; Iqbal & Sha, 2019), and learning of communication protocols between cooperative agents (Foerster et al., 2016; Sukhbaatar et al., 2016; Jiang & Lu, 2018).

**Exploration in MARL** While the fields of exploration in RL and multi-agent RL are popular, relatively little work has been done at the intersection of both. Carmel & Markovitch (1997) consider exploration with respect to opponent strategies in competitive games, and Verbeeck et al. (2005) consider exploration of a large joint action space in a load balancing problem. Jaques et al. (2018) define an intrinsic reward function for multi-agent reinforcement learning that encourages agents to take actions which have the biggest effect on other agents' behavior, otherwise referred to as "social influence." Agogino & Tumer (2008) Define metrics for evaluating the efficacy of reward functions in multi-agent domains. These works, while important, do not address the problem of exploring a large state space, and whether this exploration can be improved in multi-agent systems. A recent approach to collaborative evolutionary reinforcement learning (Khadka et al., 2019) shares some similarities with our approach. As in our work, the authors devise a method for learning a population of diverse policies with a shared replay buffer and dynamically selecting the best learner;

however, their work is focused on single-agent tasks and does not incorporate any notion of intrinsic rewards. As such, this work is not applicable to sparse reward problems in MARL.

## 3 BACKGROUND

**Dec-POMDPs** In this work, we consider the setting of decentralized POMDPs (Oliehoek et al., 2016), which are used to describe cooperative multi-agent tasks. A decentralized POMDP (Dec-POMDP) is defined by a tuple: $(\mathbf{S}, \mathbf{A}, T, \mathbf{O}, O, R, n, \gamma)$. In this setting we have $n$ total agents. $\mathbf{S}$ is the set of global states in the environment, while $\mathbf{O} = \otimes_{i \in \{1...n\}} \mathbf{O}_i$ is the set of joint observations for each agent and $\mathbf{A} = \otimes_{i \in \{1...n\}} \mathbf{A}_i$ is the set of possible joint actions for each agent. A specific joint action at one time step is denoted as $\mathbf{a} = \{a_1, \ldots, a_n\} \in \mathbf{A}$ and a joint observation is $\mathbf{o} = \{o_1, \ldots, o_n\} \in \mathbf{O}$. $T$ is the state transition function which defines the probability $P(s'|s, \mathbf{a})$, and $O$ is the observation function which defines the probability $P(\mathbf{o}|\mathbf{a}, s')$. $R$ is the reward function which maps the combination of state and joint actions to a single scalar reward. Importantly, this reward is shared between all agents, so Dec-POMDPs always describe cooperative problems. Finally, $\gamma$ is the discount factor which determines how much the agents should favor immediate reward over long-term gain.

**Soft Actor-Critic** Our approach uses Soft Actor-Critic (SAC) (Haarnoja et al., 2018) as its underlying algorithm. SAC incorporates an entropy term in the loss functions for both the actor and critic, in order to encourage exploration and prevent premature convergence to a sub-optimal deterministic policy. The policy gradient with an entropy term is computed as follows:

$$\nabla_\theta J(\pi_\theta) = \mathbb{E}_{s \sim D, a \sim \pi} \left[ \nabla_\theta \log \pi_\theta(a|s) \left( -\frac{\log \pi_\theta(a|s)}{\alpha} + Q_\psi(s, a) - b(s) \right) \right] \quad (1)$$

where $D$ is a replay buffer that stores past environment transitions, $\psi$ are the parameters of the learned critic, $b(s)$ is a state dependent baseline (e.g. the state value function $V(s)$), and $\alpha$ is a reward scale parameter determining the amount of entropy in an optimal policy. The critic is learned with the following loss function:

$$\mathcal{L}_Q(\psi) = \mathbb{E}_{(s,a,r,s') \sim D} \left[ (Q_\psi(s, a) - y)^2 \right] \quad (2)$$

$$y = r(s, a) + \gamma \mathbb{E}_{a' \sim \pi(s')} \left[ Q_{\bar\psi}(s', a') - \frac{\log(\pi_{\bar\theta}(a'|s'))}{\alpha} \right] \quad (3)$$

where $\bar\psi$ are the parameters of the target critic which is an exponential moving average of the past critics, updated as: $\bar\psi \leftarrow (1 - \tau)\bar\psi + \tau\psi$, and $\tau$ is a hyperparameter that controls the update rate.

**Centralized Training with Decentralized Execution** A number of works in deep multi-agent reinforcement learning have followed the paradigm of centralized training with decentralized execution (Lowe et al., 2017; Foerster et al., 2018; Sunehag et al., 2018; Rashid et al., 2018; Iqbal & Sha, 2019). This paradigm allows for agents to train while sharing information (or incorporating information that is unavailable at test time) but act using only local information, without requiring communication which may be costly at execution time. Since most reinforcement learning applications use simulation for training, communication between agents during the training phase has a relatively lower cost.

## 4 INTRINSIC REWARD FUNCTIONS FOR MULTI-AGENT EXPLORATION

In this section we present a set of intrinsic reward functions for exploration that incorporate information about what other agents have explored. These rewards assume that each agent (indexed by $i$) has a novelty function $f_i$ that determines how novel an observation is to it, based on its past experience. This function can be an inverse state visit count in discrete domains, or, in large/continuous domains, it can be represented by recent approaches for developing novelty-based intrinsic rewards in complex domains, such as random network distillation (Burda et al., 2019). Note that we assume that all agents share the same observation space so that each agent's novelty function can operate on all other agents' observations.

Table 1: Multi-agent intrinsic rewards for agent $i$, with $\mu(o_i) = \frac{1}{n} \sum_j f_j(o_i)$

| INDEPENDENT | MINIMUM | COVERING | BURROWING | LEADER-FOLLOWER |
|:---:|:---:|:---:|:---:|:---:|
| $f_i(o_i)$ | $\min\limits_{j \in \{1 \dots n\}} f_j(o_i)$ | $f_i(o_i)\mathbb{1}\left[f_i(o_i) > \mu(o_i)\right]$ | $f_i(o_i)\mathbb{1}\left[f_i(o_i) < \mu(o_i)\right]$ | See text |

In Table 1 we define the intrinsic rewards that we use in our experiments. INDEPENDENT rewards are analogous to single-agent approaches to exploration which define the intrinsic reward for an agent as the novelty of their new and own observation that occurs as a result of an action. The remainder of intrinsic reward functions that we consider use the novelty functions of other agents, in addition to their own, to further inform their exploration.

MINIMUM rewards consider how novel all agents find a specific agent's observation and rewards that agent based on the minimum of these novelties. This method leads to agents only being rewarded for exploring areas that no other agent has explored, which could be advantageous in scenarios where redundancy in exploration is not useful or even harmful. COVERING rewards agents for exploring areas that it considers more novel than the average agent. This reward results in agents shifting around the state space, only exploring regions as long as they are more novel to them than their average teammate. BURROWING rewards do the opposite, only rewarding agents for exploring areas that it considers less novel than the average agent. While seemingly counterintuitive, these rewards encourage agents to further explore areas they have already explored with the hope that they will discover new regions that few or no other agents have seen, which they will then consider less novel than average and continue to explore. As such, these rewards result in agents continuing to explore until they exhaust all possible intrinsic rewards from a given region (i.e. hit a dead end), somewhat akin to a depth-first search. LEADER-FOLLOWER uses burrowing rewards for the first agent, and covering rewards for the rest of the agents. This leads to an agent exploring a space thoroughly, and the rest of the agents following along and trying to cover that space.

Note that these are not meant to be a comprehensive set of intrinsic reward functions applicable to all cooperative multi-agent tasks but rather a set of examples of how exploration can be centralized in order to take other agents into account. Our approach, described in the following sections, is agnostic to the type of intrinsic rewards used and, as such, can incorporate other reward types not described here, as long as they can be computed off-policy.

## 5 LEARNING POLICIES FOR MULTI-AGENT EXPLORATION

For many tasks, it is impossible to know a priori which intrinsic rewards will be the most helpful one. Furthermore, the type of reward that is most helpful could change over the course of training if the task is sufficiently complex. In this section we present our approach for simultaneously learning policies trained with different types of intrinsic rewards and dynamically selecting the best one.

**Simultaneous Policy Learning** In order to learn policies for various types of intrinsic rewards in parallel, we utilize a shared replay buffer and off-policy learning to maximize sample efficiency. In other words, we learn policies and value functions for *all* intrinsic reward types from *all* collected data, regardless of which policies it was collected by. This parallel learning is made possible by the fact that we can compute our novelty functions off-policy, given the observations for each agent after each environment transition, which are saved in a replay buffer. For each type of reward, we learn a different "head" for our policies and critics. In other words, we learn a single network for each agent's set of policies that shares early layers and branches out into different heads for each reward type. For critics, we learn a single network across all agents that shares early layers and branches out into separate heads for each agent and reward type. We learn separate heads for intrinsic and extrinsic rewards, as in Burda et al. (2019). We provide a diagram of our model architecture in Figure 1.

We index agents by $i \in \{1 \dots n\}$ and intrinsic reward types by $j \in \{1 \dots m\}$ where $m$ is the total number of intrinsic reward types that we are considering. The policy for agent $i$, trained using reward $j$ (in addition to extrinsic rewards), is represented by $\pi_i^j$. It takes as input agent $i$'s observation, $o_i$, and outputs a distribution from which we can sample the action $a_i$. The parameters of this policy

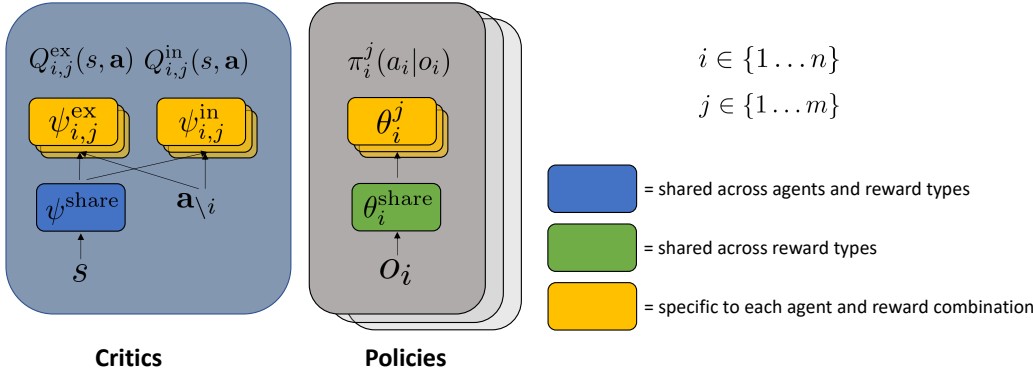

Figure 1: Diagram of our model architecture, showing how parameters for actors and critics are shared. $i$ indexes agents, while $j$ indexes reward types.

are $\Theta_i^j = \{\theta_i^{\text{share}}, \theta_i^j\}$, where $\theta_i^{\text{share}}$ is a shared base/input (for agent $i$) in a neural network and $\theta_i^j$ is a head/output specific to this reward type.

The extrinsic critic for policy head $\pi_i^j$ is represented by $Q_{i,j}^{\text{ex}}$. It takes as input the global state $s$ and the actions of all other agents $\mathbf{a}_{\setminus i}$, and it outputs the expected returns under policy $\pi_i^j$ for each possible action that agent $i$ can take, given all other agents' actions. The parameters of this critic are $\Psi_{i,j}^{\text{ex}} = \{\psi^{\text{share}}, \psi_{i,j}^{\text{ex}}\}$ where $\psi^{\text{share}}$ is a shared base across all agents and reward types. A critic with similar structure exists for predicting the intrinsic returns of actions taken by $\pi_i^j$, represented by $Q_{i,j}^{\text{in}}$, which uses the parameters: $\Psi_{i,j}^{\text{in}} = \{\psi^{\text{share}}, \psi_{i,j}^{\text{in}}\}$. Note that the intrinsic critics share the same base parameters $\psi^{\text{share}}$.

We remove the symbols representing the parameters of the policies ($\Theta$) and the critics ($\Psi$) for readability. In our notation we use the absence of a subscript or superscript to refer to a group. For example $\pi^j$, refers to *all* agents' policies trained on intrinsic reward $j$. We train our critics with the following loss function, adapted from soft actor-critic:

$$\mathcal{L}_Q(\Psi) = \mathbb{E}_{(s,\mathbf{o},\mathbf{a},r,s',\mathbf{o}')\sim D}\left[\sum_{j=1}^m \sum_{i=1}^n (Q_{i,j}^{\text{ex}}(s,\mathbf{a}) - y_{i,j}^{\text{ex}})^2 + (Q_{i,j}^{\text{in}}(s,\mathbf{a}) - y_{i,j}^{\text{in}})^2\right] \quad (4)$$

$$y_{i,j}^{\text{ex}} = r^{\text{ex}}(s,\mathbf{a}) + \gamma \mathbb{E}_{\mathbf{a}'\sim\bar{\pi}^j(\mathbf{o}')}\left[\bar{Q}_{i,j}^{\text{ex}}(s',\mathbf{a}') - \frac{\log(\bar{\pi}_i^j(a_i'|o_i'))}{\alpha}\right] \quad (5)$$

$$y_{i,j}^{\text{in}} = r_{i,j}^{\text{in}}(o_i') + \gamma \mathbb{E}_{\mathbf{a}'\sim\bar{\pi}^j(\mathbf{o}')}\left[\bar{Q}_{i,j}^{\text{in}}(s',\mathbf{a}') - \frac{\log(\bar{\pi}_i^j(a_i'|o_i'))}{\alpha}\right] \quad (6)$$

where $\bar{Q}$ refers to the target Q-function, an exponential weighted average of the past Q-functions, used for stability, and $\bar{\pi}$ are similarly updated target policies. The intrinsic rewards laid out in Table 1 are represented as a function of the observations that results from the action taken, $r_{i,j}^{\text{in}}(o_i')$ where $j$ specifies the type of reward. Importantly, we can calculate these loss functions for expected intrinsic and extrinsic returns for all policies given a single environment transition, allowing us to learn multiple policies for each agent in parallel. We train each policy head with the following

gradient:

$$\nabla_{\Theta_i^j} J(\pi_i^j) = \mathbb{E}_{(s,\mathbf{o})\sim D, \mathbf{a}\sim\pi^j} \left[ \nabla_{\Theta_i^j} \log \pi_i^j(a_i|o_i) \left( -\frac{\log \pi_i^j(a_i|o_i)}{\alpha} + A_i^j(s,\mathbf{a}) \right) \right] \tag{7}$$

$$A_i^j(s,\mathbf{a}) = Q_{i,j}^{\text{ex}}(s,\mathbf{a}) + \beta Q_{i,j}^{\text{in}}(s,\mathbf{a}) - V_i^j(s) \tag{8}$$

$$V_i^j(s) = \sum_{a_i' \in \mathbf{A}_i} \pi_i^j(a_i'|o_i)(Q_{i,j}^{\text{ex}}(s, \{a_i', \mathbf{a}_{\backslash i}\}) + \beta Q_{i,j}^{\text{in}}(s, \{a_i', \mathbf{a}_{\backslash i}\})) \tag{9}$$

where $\beta$ is a scalar that determines the weight of the intrinsic rewards, relative to extrinsic rewards, and $A_i^j$ is a multi-agent advantage function (Foerster et al., 2018; Iqbal & Sha, 2019), used for helping with multi-agent credit assignment.

**Dynamic Policy Selection**  Now that we have established a method for simultaneously learning policies using different intrinsic reward types, we must devise a means of selecting between these policies. In order to select policies to use for environment rollouts, we must consider which policies maximize extrinsic returns, while taking into account the fact that there may still be "unknown unknowns," or regions that the agents have not seen yet where they may be able to further increase their extrinsic returns. As such, we must learn a meta-policy that, at the beginning of each episode, selects between the different sets of policies trained on different intrinsic rewards and maximizes extrinsic returns without collapsing to a single set of policies too early. We parameterized the selector policy $\Pi$ with a vector, $\phi$, that contains an entry for every reward type. The probability of sampling head $j$ is: $\Pi(j) \propto \exp(\phi[j])$. Unlike the action policies, this high-level policy does not take any inputs, a we simply want to learn which set of policies trained on the individual intrinsic reward functions has the highest expected extrinsic returns from the beginning of the episode.

The most sensible metric for selecting policies is the expected extrinsic returns given by each policy head. We can use policy gradients to train the policy selector, $\Pi$, to maximize this value using the returns received when performing rollouts in the environment. We use the following gradient to train $\Pi$:

$$\nabla_\phi J(\Pi) = \mathbb{E}_{h\sim\Pi} \left[ \nabla_\phi \log \Pi(h) \left( -\frac{\log \Pi(h)}{\eta} + R_h^{\text{ex}} - b_\Pi \right) \right] \tag{10}$$

$$R_h^{\text{ex}} = \sum_{t=0}^{T} \gamma^t r^{\text{ex}}(s_t, \mathbf{a}_t)|\mathbf{a} \sim \pi^h(\mathbf{o}_t), \quad b_\Pi = \sum_{h'}^{m} \Pi(h')\mu_{h'} \tag{11}$$

where $\mu_h$ is a running mean of the returns received by head $h$ in the past, and $\eta$ is a parameter similar to $\alpha$ for the low-level policies, which promotes entropy in the selector policy. Entropy in the policy selector is important in order to prevent it from collapsing onto a single exploration type that does well at first but does not continue to explore as effectively as others. As such, we can learn a diverse set of behaviors based on various multi-agent intrinsic reward functions and select the one that maximizes performance on the task at hand at any point during training, while continuing to consider other policies that may lead to greater rewards.

## 6  EXPERIMENTS

We begin by describing our evaluation domains and then present experimental results which demonstrate the effectiveness of our approach. We provide additional details in the appendix and will share code for both the model and environments.

We use a maximum of four agents in gridworld and two agents in VizDoom. We encode several tasks in both domains related to collecting the items (displayed in yellow in Figure 2) which each require different types of exploration: **TASK 1** Agents must cooperatively collect *all* treasure on the map in order to complete the task; **TASK 2** Agents must all collect the *same* treasure. The first agent to collect a treasure during an episode determines the goal for the rest of the agents. **TASK 3** Agents must all collect the specific treasure that is assigned to them. The two agent version of each task uses agents 1-2 and treasure A-B, while the three agent versions use 1-3, A-C, and the four agent versions use 1-4, A-D. Agents receive a negative time penalty towards extrinsic rewards

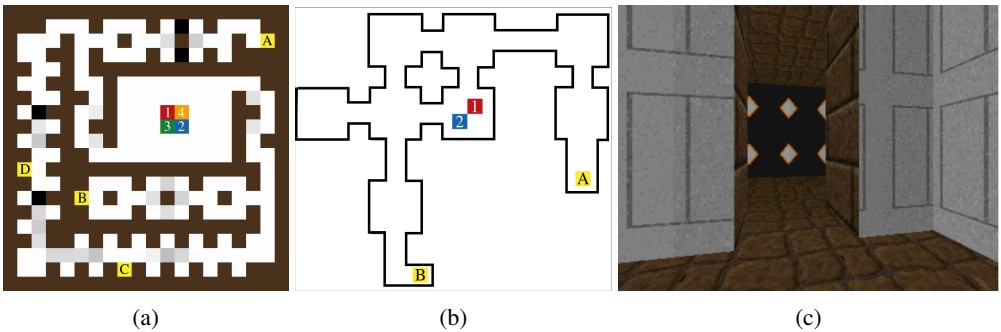

Figure 2: (Left) Rendering of our gridworld domain. Agents start each episode in the central room and must complete various tasks related to collecting the yellow treasures placed around the map. (Center) Top-Down view of VizDoom "My Way Home" map, modified for multi-agent experiments (Right) Egocentric view in VizDoom used for agents' observations

at each step, so they are motivated to complete the task as quickly as possible. The only positive extrinsic reward comes from any agent collecting a treasure that is allowed by the specific task, and rewards are shared between all agents. The optimal strategy in **TASK 1** is for agents to spread out and explore separate portions of the map, while in **TASK 2** they should explore the same areas, and in **TASK 3** they should explore independently.

## 6.1 GRIDWORLD DOMAIN

We first test our approach using a multi-agent gridworld domain (pictured in Fig. 2a), which allows us to design environments where the primary challenge lies in a combination of exploring the state space efficiently and coordinating behaviors.

The environment includes two sources of stochasticity: random transitions and black holes. At each step there is a 10% chance of an agent's action being replaced by a random one. Furthermore, there are several "black holes" placed around the map which have a probability of opening at each time step. This probability changes at each step using a biased random walk such that it moves toward one, until the hole opens and it resets to zero. If an agent steps into a black hole when it is open, they will be sent back to their starting position. The spaces colored as black are holes that are currently open, while the gray spaces are holes that have the possibility of opening at the next step (the darker they are, the higher the probability). We set the rate of black holes dropping out to be higher in **TASK 1** than the other 2 tasks, in order to balance the difficulty.

The novelty function for each agent $f_i$, which is used for calculating the intrinsic rewards in Table 1, is defined as $\frac{1}{N^\zeta}$, where $N$ is the number of times that the agent has visited its current cell and $\zeta$ is a decay rate selected as a hyperparameter (we find that $\zeta = 0.7$ works well for our purposes).

## 6.2 VIZDOOM DOMAIN

In order to test our method's ability to scale to more complex environments with similarly challenging exploration tasks, we implement tasks analogous to those in our gridworld environment (i.e. extrinsic rewards are defined identically) in the VizDoom framework (Kempka et al., 2016). We use the "My Way Home" map, which has been used as a test bed for single agent exploration techniques (Pathak et al., 2017), and modify it for multi-agent tasks (pictured in Figure 2b). Since the agents are moved to a central location closer to their rewards than in the original map, we lower the action repeat from 4 to 2, in order to force agents to take twice as many steps in order to explore the same areas, maintaining the challenging nature of exploration in the original task.

As in the gridworld setting, we use count-based intrinsic rewards for VizDoom; however, since VizDoom is not a discrete domain, we separate agents' $(x, y)$ positions into discrete bins and use the counts for these bins. We again find that $\zeta = 0.7$ to work well in our experiments.

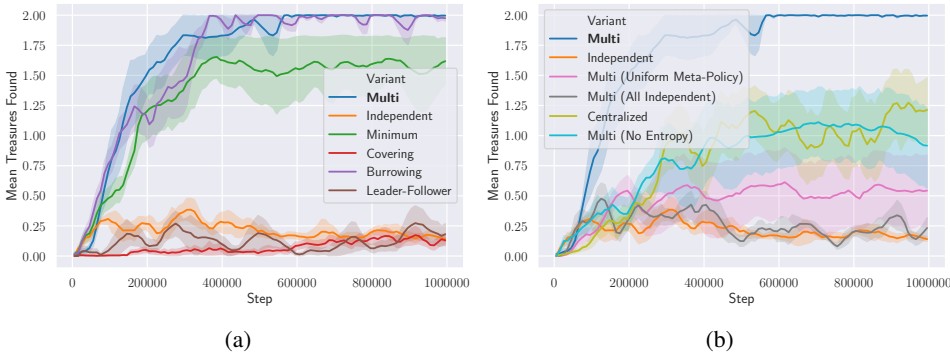

(a)                                                          (b)

Figure 3: (Left) Mean number of trasures found per episode on TASK 1 with 2 agents in the gridworld domain. Shaded region is a 68% confidence interval across 6 runs of the running mean over the past 100 episodes. Our approach (MULTI-EXPLORATION) is competitive with the best individual intrinsic reward function, using the same number of environment samples without any prior knowledge provided. (Right) Ablations of our model in the same setting. We show that both aspects of our approach (the meta-policy selector and the diverse intrinsic reward functions) are crucial for successful completion of exploration tasks requiring coordination.

## 6.3 MAIN RESULTS

Figure 3a demonstrates the results of our approach over the course of training on the 2 agent version of **TASK 1** in gridworld, and the final results on each task/agent/domain combination can be found in Table 2. The full training curves for all settings can be found in the appendix (Section A.4). We train a team of agents using each of the multi-agent intrinsic reward functions defined in Table 1 individually, and then test our dynamic policy selection approach. We find that our approach is competitive with, or outperforms the best performing individual exploration method in nearly all tasks. This performance is exciting since our method receives no prior information about which type of exploration would work best, while each type carries its own inductive bias. Notably our learned policy selector learns to select the policies trained on intrinsic rewards that do well individually on the tasks. For instance, on **TASK 1** with 2 agents, we find that our policy selector consistently selects BURROWING and MINIMUM rewards, the two best performing reward functions on that task. Furthermore, we find that our results on the more complex VizDoom domain mirror those in the gridworld, indicating that our methods are not limited to discrete domains, assuming that a reliable way for measuring the novelty of observations exists.

Interestingly, our approach is sometimes able to significantly surpass the performance of the best individual reward function on **TASK 3**. This task requires agents to collect the specific reward assigned to them, so we expect independent exploration to be the most effective; however, exploration types that perform "divide-and-conquer" type behavior such as BURROWING and MINIMUM have the potential to drastically speed up the exploration process if they happen to divide the space correctly, leading to a stark success-failure contrast in runs of these types. Since our method MULTI can select policies trained on these rewards, and otherwise fall back on INDEPENDENT policies if they are not working, we find that our method is able to surpass all individual reward types.

We find that our approach is unable to match the performance of the best individual method on **TASK 2** in some settings (gridworld with 3 agents and VizDoom). This lack of success may be an indication that these particular settings require commitment to a specific exploration strategy early on in training, highlighting a limitation of our approach. Our method requires testing out all policies until we find one that reaches high extrinsic rewards, which can dilute the effectiveness of exploration early on.

## 6.4 ANALYSIS

**Characteristics of Different Intrinsic Rewards** In order to better understand how each reward type encourages agents to explore the state space, we visualize their exploration in videos, viewable at the anonymized link below.[1]. INDEPENDENT rewards, as expected, result in agents exploring the whole state space without taking other agents into consideration. As a result, on **TASK 1**, which

---

[1]https://sites.google.com/view/multi-exploration-iclr2020/home

Table 2: # of treasures found with standard deviation across 6 runs. Scores where the best mean score falls within one standard deviation of the score distribution are bolded.

| Task | $n$ | **GRIDWORLD** | | | | | |
|---|---|---|---|---|---|---|---|
| | | **Intrinsic reward type (fixed or adaptive as in our approach MULTI)** | | | | | |
| | | INDEPENDENT | MINIMUM | COVERING | BURROWING | LEAD-FOLLOW | MULTI |
| 1 | 2 | $0.14 \pm 0.05$ | $\mathbf{1.62 \pm 0.59}$ | $0.13 \pm 0.12$ | $\mathbf{1.98 \pm 0.06}$ | $0.18 \pm 0.24$ | $\mathbf{2.00 \pm 0.00}$ |
| | 3 | $1.16 \pm 0.11$ | $\mathbf{1.49 \pm 0.76}$ | $0.00 \pm 0.00$ | $\mathbf{2.06 \pm 1.05}$ | $0.34 \pm 0.45$ | $\mathbf{2.23 \pm 0.73}$ |
| | 4 | $0.84 \pm 0.29$ | $\mathbf{1.78 \pm 0.44}$ | $0.00 \pm 0.00$ | $\mathbf{1.90 \pm 0.49}$ | $1.17 \pm 0.39$ | $\mathbf{2.04 \pm 0.61}$ |
| 2 | 2 | $\mathbf{2.00 \pm 0.00}$ | $0.92 \pm 0.10$ | $\mathbf{1.11 \pm 0.99}$ | $0.98 \pm 0.05$ | $\mathbf{1.73 \pm 0.66}$ | $\mathbf{1.83 \pm 0.41}$ |
| | 3 | $\mathbf{2.66 \pm 0.80}$ | $1.11 \pm 0.29$ | $0.54 \pm 0.80$ | $1.80 \pm 0.29$ | $\mathbf{3.00 \pm 0.00}$ | $1.80 \pm 0.71$ |
| | 4 | $\mathbf{1.83 \pm 1.08}$ | $0.93 \pm 0.13$ | $0.22 \pm 0.18$ | $\mathbf{1.99 \pm 0.67}$ | $\mathbf{2.66 \pm 2.06}$ | $\mathbf{2.54 \pm 1.21}$ |
| 3 | 2 | $\mathbf{1.39 \pm 0.94}$ | $0.67 \pm 1.03$ | $0.29 \pm 0.37$ | $0.67 \pm 1.03$ | $0.83 \pm 0.67$ | $\mathbf{2.00 \pm 0.00}$ |
| | 3 | $\mathbf{1.68 \pm 0.70}$ | $0.60 \pm 0.73$ | $0.09 \pm 0.08$ | $\mathbf{1.35 \pm 1.16}$ | $1.59 \pm 0.83$ | $\mathbf{2.21 \pm 0.91}$ |
| | 4 | $1.12 \pm 0.47$ | $1.36 \pm 0.71$ | $0.05 \pm 0.05$ | $\mathbf{2.14 \pm 1.49}$ | $0.68 \pm 0.53$ | $\mathbf{1.73 \pm 0.47}$ |
| | | **VIZDOOM** | | | | | |
| 1 | 2 | $0.94 \pm 0.54$ | $\mathbf{1.57 \pm 0.74}$ | $0.16 \pm 0.17$ | $\mathbf{1.94 \pm 0.10}$ | $0.61 \pm 0.43$ | $\mathbf{1.98 \pm 0.03}$ |
| 2 | | $\mathbf{1.52 \pm 0.75}$ | $\mathbf{1.53 \pm 0.74}$ | $0.70 \pm 1.00$ | $0.63 \pm 0.04$ | $\mathbf{1.93 \pm 0.10}$ | $1.23 \pm 0.65$ |
| 3 | | $0.18 \pm 0.19$ | $\mathbf{0.64 \pm 1.05}$ | $0.45 \pm 0.46$ | $0.29 \pm 0.25$ | $0.20 \pm 0.17$ | $\mathbf{1.64 \pm 0.63}$ |

requires coordination between agents to spread out and explore different areas, INDEPENDENT rewards struggle; however, on **TASK 3**, where agents receive individualized goals, independent exploration usually performs better, relative to the other methods. **TASK 2** also requires coordination, but the rate of black holes dropping out in the gridworld version is lower on that task, making exploration easier. As a result, INDEPENDENT rewards perform well on **TASK 2**; however, we find that LEADER-FOLLOWER also performs well on this task, expecially when more agents are involved, indicating that these rewards do a good job of biasing agents toward exploring similar regions of the environment.

MIMIMUM rewards prevent agents from exploring the same regions redundantly but can lead to situations where one of the agents is the first to explore all regions that provide sparse extrinsic rewards. In these cases, other agents are not aware of the extrinsic rewards and are also not motivated to explore for them since another agent has already done so. COVERING rewards, as expected, lead to behavior where agents are constantly switching up the regions that they explore. While this behavior does not prove to be useful in the tasks we test since the switching slows down overall exploration progress, it may be useful in scenarios where agents are required to spread out. Finally, BURROWING rewards cause agents to each explore different subregions and continue to explore those regions until they exhaust their options. This behavior is particularly effective on **TASK 1**, where agents are best served by spreading out and exploring the whole map in a mutually exclusive fashion.

**Ablations** We compare to a baseline meta-policy which simply selects the action policies uniformly at random. We find that our approach is significantly superior to this baseline (see Figure 3b *Multi (Uniform Meta-Policy)*). Furthermore, we test a version of our method where all policies (with different random initializations) are trained on independent rewards (*Multi (All Independent)*). The purpose of this ablation is to test the degree to which the specific multi-agent intrinsic reward functions are helpful, as opposed to simply providing multiple options at each episode. Again, we find that our method outperforms the baseline, indicating that both aspects of our approach (diverse intrinsic reward functions which share information across agents, and a meta-policy selector that maximizes extrinsic rewards) are crucial for success in multi-agent exploration tasks.

We perform two further ablations/comparisons. Results on task 1 with 2 agents in GridWorld are viewable in Figure 3b, and results on tasks 2 and 3 with 2 agents are viewable in the Appendix (A.5). In the first (*Centralized*) we compute intrinsic rewards under the assumption that all agents are treated as one agent. In other words, we use the inverse count of the number of times that **all** agents have jointly taken up their combined positions, rather than considering agents independently. While this reward function will ensure that the global state space is thoroughly searched, it lacks the inductive biases toward spatial coordination that our reward functions incorporate. As such, it does

not learn as efficiently as our method. In the second (*Multi (No Entropy)*) we remove the entropy term from the head selector loss function in order to test its importance. We find that this ablation is unable to match the performance of the full method, indicating that entropy is crucial in making sure that our method does not converge early to a suboptimal policy selector.

## 7 Conclusion

We propose a set of multi-agent intrinsic reward functions with differing properties, and compare them both qualitatively (through videos) and quantitatively on several multi-agent exploration tasks in both a gridworld domain as well as in VizDoom. Overall, we can see that cooperative multi-agent tasks can, in many cases, benefit from intrinsic rewards that take into account what other agents have explored, but there are various ways to incorporate that information, each with differing properties. As such, we propose a method for learning policies for all intrinsic reward types simultaneously while dynamically selecting the most effective ones. We show that our method is capable of matching or surpassing the performance of the best performing intrinsic reward type on various tasks while using the same number of samples collected from the environment. In future work we hope to introduce methods for directly learning the multi-agent intrinsic reward functions, rather than selecting from a set.

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

# A APPENDIX

## A.1 ENVIRONMENT DETAILS

### A.1.1 GRIDWORLD

The black holes which send agents back to their starting positions if they are stepped into are an important aspect of the environment, as they add difficulty to exploration. The probability, $\rho$, of a black hole opening at each step, $t$, evolves as such: $\rho_{t+1} = \rho_t + \mathcal{N}(\mu, \sigma)$, where $\mu = \sigma = 0.05$ for **TASK 1** and $\mu = \sigma = 0.005$ for **2** and **3**.

Agents observe their global position in $(x, y)$ coordinates (scalars), as well as local information regarding walls in adjacent spaces, the probability of their adjacent spaces opening into a black hole, the relative position of other agents (if they are within 3 spaces), as well as information about which treasures the agent has already collected in the given episode. The global state is represented by the $(x, y)$ coordinates of all agents, as one-hot encoded vectors for $x$ and $y$ separately, as well as the local information of all agents regarding black holes, walls, and treasures collected. Each agent's action space consists of the 4 cardinal directions as well as an option to not move, which is helpful in cases where an agent is waiting for a black hole to be safe to cross.

### A.1.2 VIZDOOM

Agents receive their egocentric view (Figure 2c) in the form of 48x48 grayscale images as observations along with an indicator of which agents (if any) have collected each reward, and we use a vector based global state which includes all agents' $(x, y)$ positions and velocities, their orientations, as well as the same indicator of which agent has collected each reward. As in the gridworld setting, we use count-based intrinsic rewards for VizDoom; however, since VizDoom is not a discrete domain, we separate agents' $(x, y)$ positions into discrete bins and use the counts for these bins. There are 30 bins in the $x$ dimension and 26 in the $y$ dimension. $(x, y)$ positions in the global state are represented both as scalars and one-hot vectors indicating which bin the agents are currently occupying. Each agent can choose from 3 actions at each time step: turn left, turn right, or go forward.

## A.2 TRAINING DETAILS

The training procedure is detailed in Algorithm 1, and all hyperparameters are listed in Tables 3 and 4. Hyperparameters were selected by tuning one parameter at a time through intuition on task 1 with 2 agents and then applying to the rest of the settings with minimal changes. Where hyperparameters differ between settings, we make a footnote denoting them as such.

---

**Algorithm 1** Training Procedure for Multi-Explore w/ Soft Actor-Critic (Haarnoja et al., 2018)

---

1: Initialize environment with $n$ agents
2: Initialize replay buffer, $D$
3: $t_{\text{update}} \leftarrow 0$
4: $t_{\text{ep}} \leftarrow$ max ep length
5: **for** $t = 1 \ldots$ total steps **do**
6:     **if** episode done or $t_{\text{ep}} ==$ max ep length **then**
7:         **for** $j = 1 \ldots$ num updates **do**
8:             UPDATESELECTOR(R, h)                 ▷ Eqs 10-11 in main text
9:         **end for**
10:         $s, \mathbf{o} \leftarrow$ RESETENV
11:         $h \sim \Pi$                              ▷ Sample policy head
12:         $t_{\text{ep}} \leftarrow 0$
13:         $R \leftarrow 0$
14:     **end if**
15:     Select actions $a_i \sim \pi_i^h(\cdot|o_i)$ for each agent, $i$
16:     Send actions to environment and get $s$, $\mathbf{o}$, $r$
17:     $R \leftarrow R + \gamma^{t_{\text{ep}}} r$
18:     Store transitions for all environments in $D$
19:     $t_{\text{update}} + = 1$
20:     $t_{\text{ep}} + = 1$
21:     **if** $t_{\text{update}} ==$ steps per update **then**
22:         **for** $j = 1 \ldots$ num updates **do**
23:             Sample minibatch, $B$
24:             UPDATECRITIC($B$)             ▷ Eqs 4-6 in main text
25:             UPDATEPOLICIES($B$)        ▷ Eqs 7-9 in main text
26:             Update target parameters:

$$\bar{\Psi} = \tau\bar{\Psi} + (1 - \tau)\Psi$$

$$\bar{\Theta} = \tau\bar{\Theta} + (1 - \tau)\Theta$$

27:         **end for**
28:         $t_{\text{update}} \leftarrow 0$
29:     **end if**
30: **end for**

---

Table 3: Hyperparameter settings across all runs in gridworld.

| Name | Description | Value |
|---|---|---|
| $Q$ lr | learning rate for centralized critic | 0.001 |
| $Q$ optimizer | optimizer for centralized critic | Adam (Kingma & Ba, 2014) |
| $\pi$ lr | learning rate for decentralized policies | 0.001 |
| $\pi$ optimizer | optimizer for decentralized policies | Adam |
| $\Pi$ lr | learning rate for policy selector | 0.04 |
| $\Pi$ optimizer | optimizer for policy selector | SGD |
| $\tau$ | target function update rate | 0.005 |
| bs | batch size | 1024 |
| total steps | number of total environment steps | $1e6$ |
| steps per update | number of environment steps between updates | 100 |
| niters | number of iterations per update | 50 |
| max ep length | maximum length of an episode before resetting | 500 |
| $\Psi$ penalty | coefficient for weight decay on parameters of Q-function | 0.001 |
| $\Theta$ penalty | coefficient on $L_2$ penalty on pre-softmax output of policies | 0.001 |
| $\theta$ penalty | coefficient for weight decay on parameters of policy selector | 0.001 |
| $|D|$ | maximum size of replay buffer | $1e6$ |
| $\alpha$ | action policy reward scale | 100 |
| $\eta$ | selector policy reward scale | 5 |
| $\gamma$ | discount factor | 0.99 |
| $\beta$ | relative weight of intrisic rewards to extrinsic | 0.1 |
| $\zeta$ | decay rate of count-based rewards | 0.7 |

Table 4: Hyperparameter settings across all runs in VizDoom (only where different from Table 3).

| Name | Description | Value |
|---|---|---|
| $Q$ lr | learning rate for centralized critic | 0.0005 |
| $\pi$ lr | learning rate for decentralized policies | 0.0005 |
| bs | batch size | 128 |
| $|D|$ | maximum size of replay buffer | $5e5$ |

## A.3 NETWORK ARCHITECTURES

In this section we list, in pseudo-code, the architectures we used for all policies and critics

### A.3.1 GRIDWORLD

$\theta_i^{\text{share}}$ (shared for policy heads):

```
obs_size = observations.shape[1]
fc1 = Linear(in_dim=obs_size, out_dim=128)
nl1 = ReLU()
```

$\theta_i^j$ (specific to each policy head):

```
n_acs = actions.shape[1]
```

```
fc2 = Linear(in_dim=fc1.out_dim, out_dim=32)
nl2 = ReLU()
fc3 = Linear(in_dim=fc2.out_sim, out_dim=n_acs)
```

$\psi^{\text{share}}$ (shared across critics for all agents and reward types):

```
state_size = states.shape[1]
fc1 = Linear(in_dim=state_size, out_dim=128)
nl1 = ReLU()
```

$\psi_{i,j}$ (specific to each agent/policy head combination, same architecture for extrinsic and intrinsic critics):

```
n_acs = actions.shape[1]
# fc2 takes other agents' actions as input
fc2 = Linear(in_dim=fc1.out_dim + (num_agents - 1) * n_acs, out_dim=128)
nl2 = ReLU()
fc3 = Linear(in_dim=fc2.out_dim, out_dim=n_acs)
```

### A.3.2 VIZDOOM

$\theta_i^{\text{share}}$ (shared for policy heads belonging to one agent):

```
# vector observation encoder
vect_obs_size = vector_observations.shape[1]
vect_fc = Linear(in_dim=obs_size, out_dim=32)
vect_nl = ReLU()
# image observation encoder
img_obs_channels = image_observations.shape[1]
pad1 = ReflectionPadding(size=1)
conv1 = Conv2D(in_channels=img_obs_channels, out_channels=32,
    filter_size=3, stride=2)
conv_nl1 = ReLU()
pad2 = ReflectionPadding(size=1)
conv2 = Conv2D(in_channels=conv1.out_channels, out_channels=32,
    filter_size=3, stride=2)
conv_nl2 = ReLU()
pad3 = ReflectionPadding(size=1)
conv3 = Conv2D(in_channels=conv2.out_channels, out_channels=32,
    filter_size=3, stride=2)
conv_nl3 = ReLU()
pad4 = ReflectionPadding(size=1)
conv4 = Conv2D(in_channels=conv3.out_channels, out_channels=32,
    filter_size=3, stride=2)
conv_nl4 = ReLU()
conv_flatten = Flatten() # flatten output of conv layers
conv_fc = Linear(in_dim=conv_flatten.out_dim, out_dim=128)
conv_fc_nl = ReLU()
```

$\theta_i^j$ (specific to each policy head):

```
n_acs = actions.shape[1]
# takes concatenation of image and vector encodings as input
fc_out1 = Linear(in_dim=conv_fc.out_dim + vect_fc.out_dim, out_dim=32)
fc_out_nl = ReLU()
fc_out2 = Linear(in_dim=fc_out1.out_dim, out_dim=n_acs)
```

$\psi^{\text{share}}$ (shared across critics for all agents and reward types):

```
state_size = states.shape[1]
fc1 = Linear(in_dim=state_size, out_dim=256)
nl1 = ReLU()
```

$\psi_{i,j}$ (specific to each agent/policy head combination, same architecture for extrinsic and intrinsic critics):

```
n_acs = actions.shape[1]
# fc2 takes other agents' actions as input
fc2 = Linear(in_dim=fc1.out_dim + (num_agents - 1) * n_acs, out_dim=256)
nl2 = ReLU()
fc3 = Linear(in_dim=fc2.out_dim, out_dim=n_acs)
```

### A.4 TRAINING CURVES

### A.4.1 GRIDWORLD

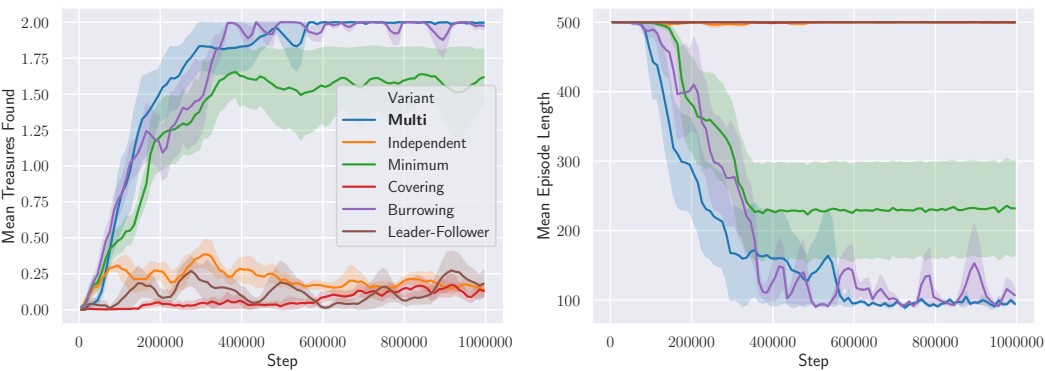

Figure 4: Results on Task 1 in Gridworld with 2 agents.

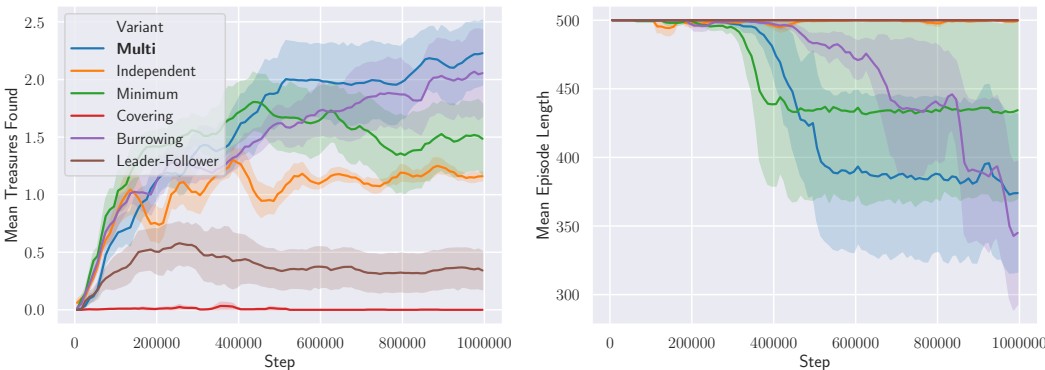

Figure 5: Results on Task 1 in Gridworld with 3 agents.

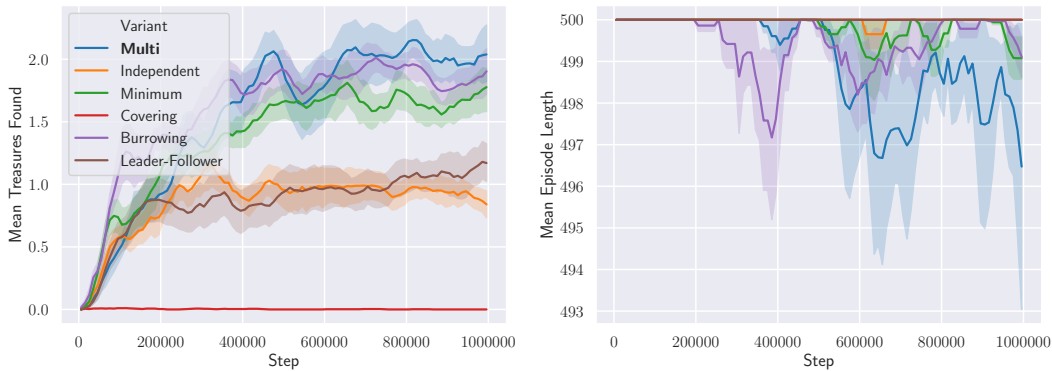

Figure 6: Results on Task 1 in Gridworld with 4 agents.

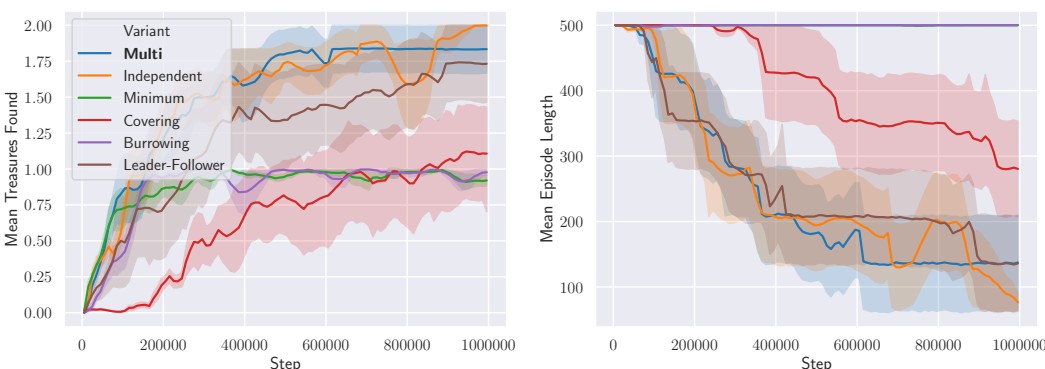

Figure 7: Results on Task 2 in Gridworld with 2 agents.

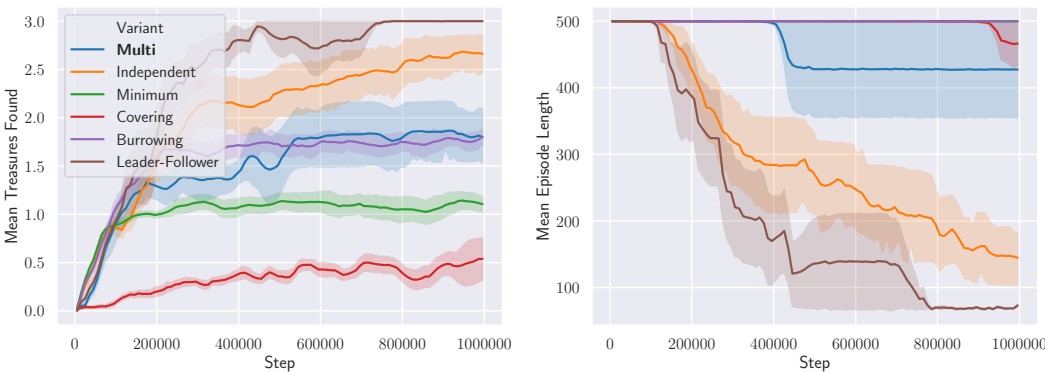

Figure 8: Results on Task 2 in Gridworld with 3 agents.

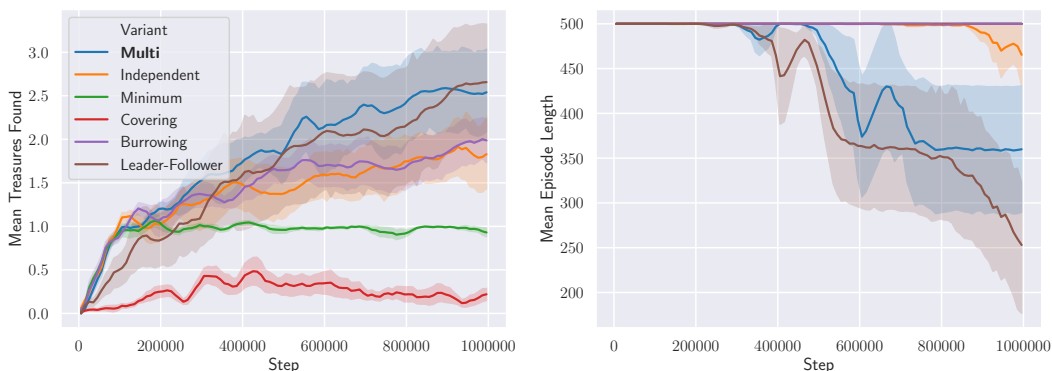

Figure 9: Results on Task 2 in Gridworld with 4 agents.

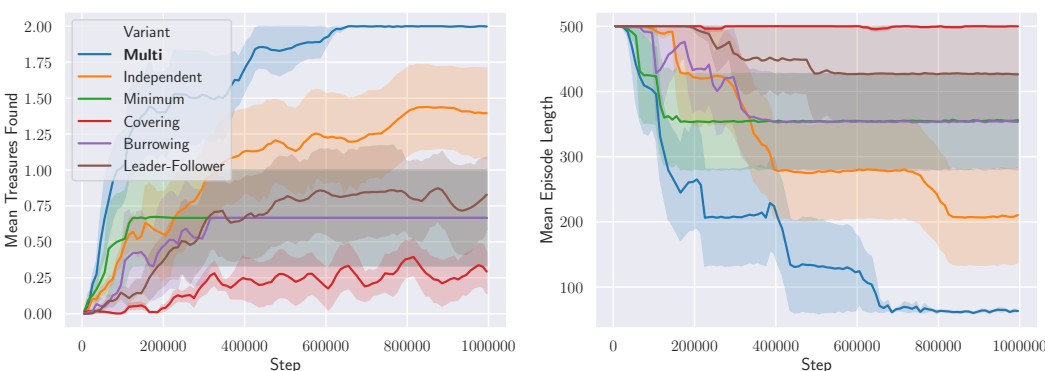

Figure 10: Results on Task 3 in Gridworld with 2 agents.

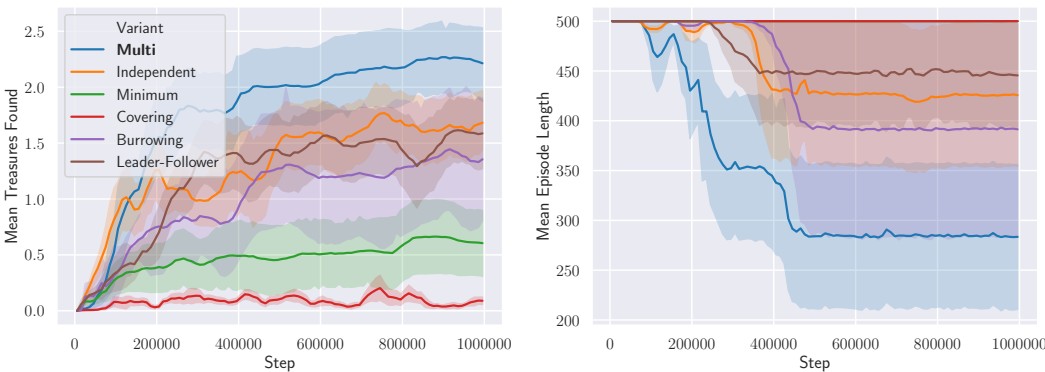

Figure 11: Results on Task 3 in Gridworld with 3 agents.

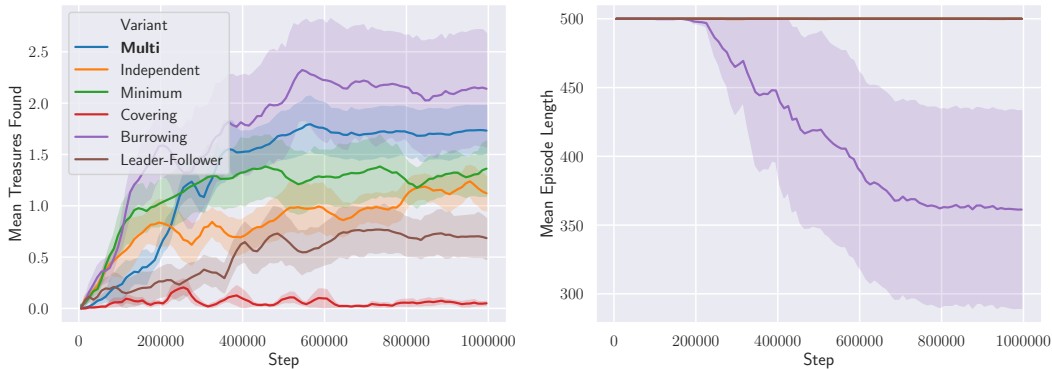

Figure 12: Results on Task 3 in Gridworld with 4 agents.

### A.4.2 VIZDOOM

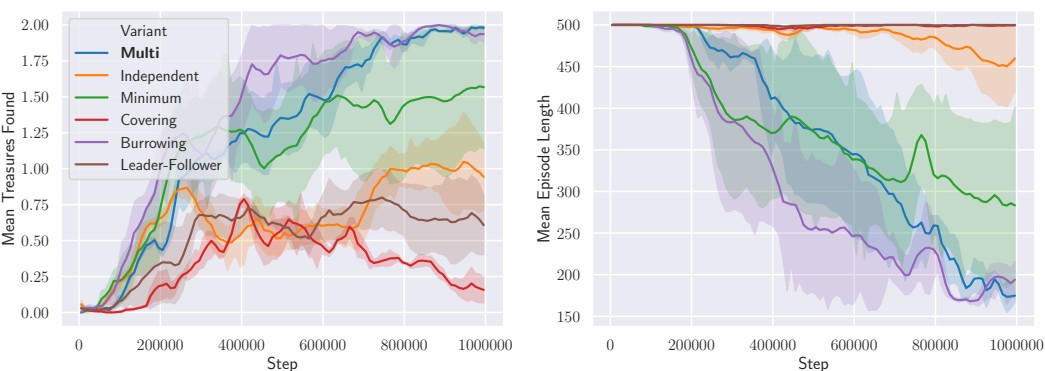

Figure 13: Results on Task 1 in Vizdoom.

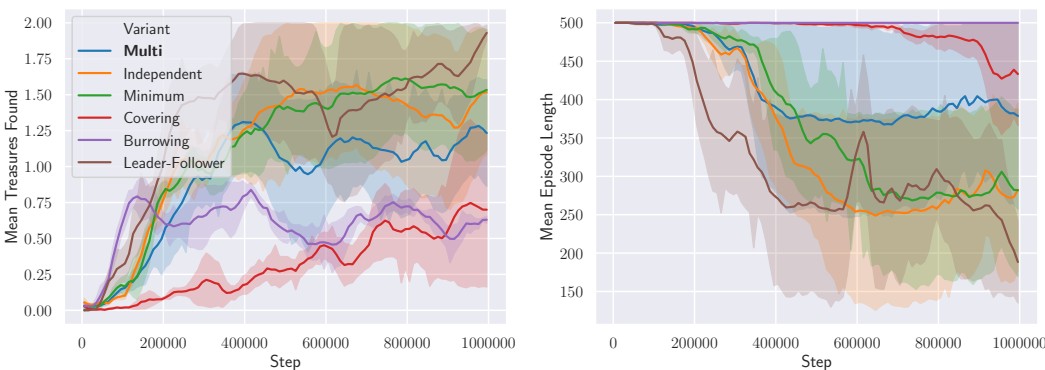

Figure 14: Results on Task 2 in Vizdoom.

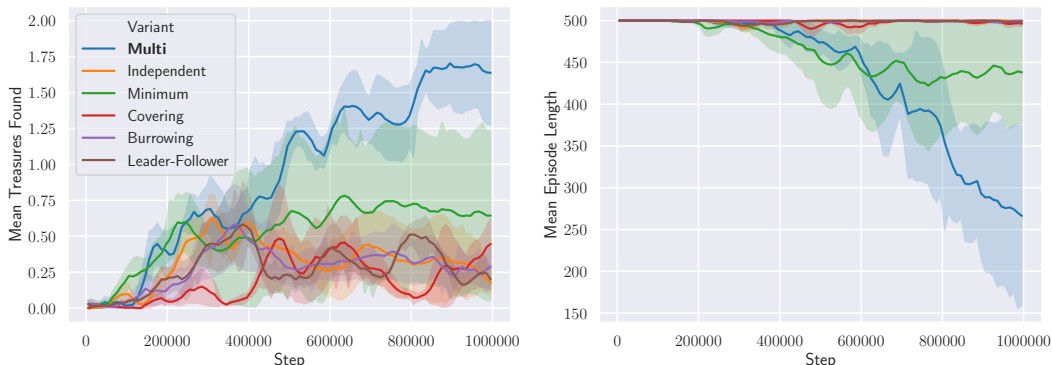

Figure 15: Results on Task 3 in Vizdoom.

### A.5 More Ablations

In this section we consider two ablations/comparisons to our model across all three tasks in the 2 agent version of gridworld. In the first (*Centralized*) we compute intrinsic rewards under the assumption that all agents are treated as one agent. In other words, we use the inverse count of the number of times that **all** agents have jointly taken up their combined positions, rather than considering agents independently. While this reward function will ensure that the global state space is thoroughly searched, it lacks the inductive biases toward spatial coordination that our reward functions incorporate. As such, it does not learn as efficiently as our method in any of the three tasks. In the second (*Multi (No Entropy)*) we remove the entropy term from the head selector loss function in order to test its importance. We find that this ablation is unable to match the performance of the full method, indicating that entropy is crucial in making sure that our method does not converge early to a suboptimal policy selector.

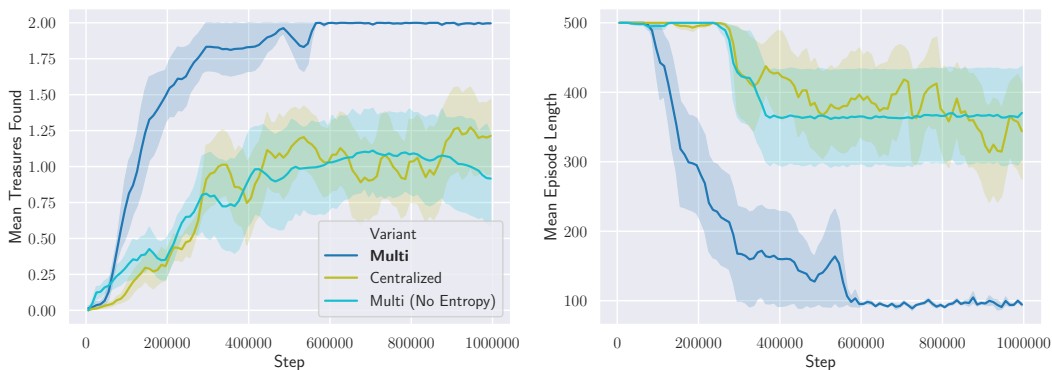

Figure 16: Ablations on Task 1 in Gridworld with 2 agents.

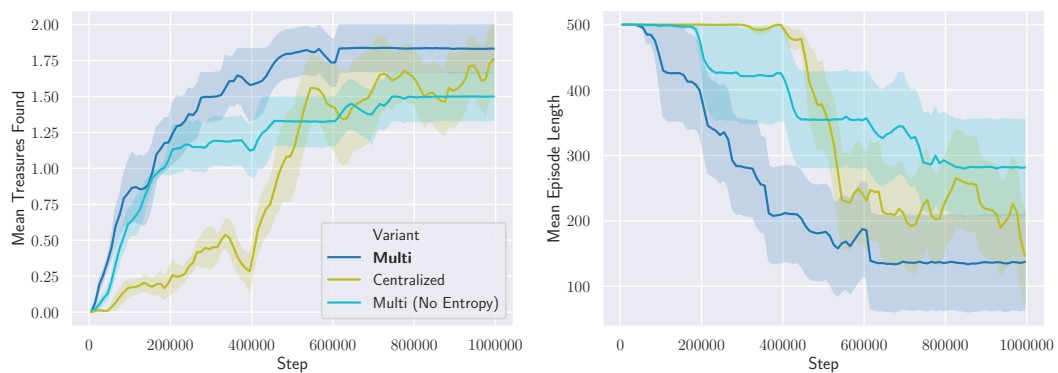

Figure 17: Ablations on Task 2 in Gridworld with 2 agents.

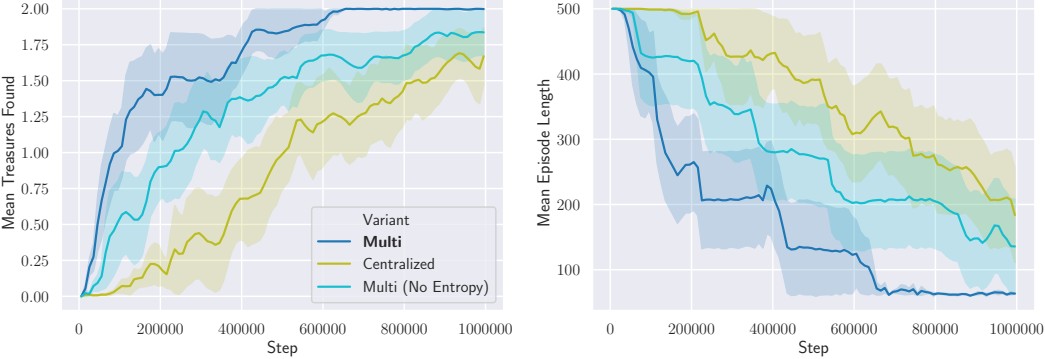

Figure 18: Ablations on Task 3 in Gridworld with 2 agents.

## A.6  ANALYZING META-POLICY

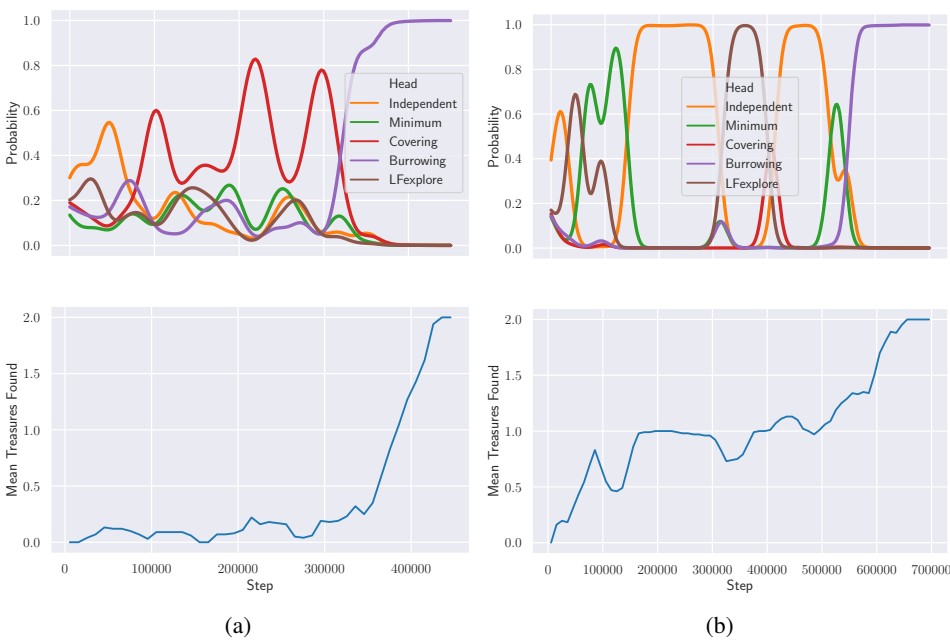

Figure 19: Two runs of our method on GridWorld Task 3 with 2 agents. Top row shows the evolution of the meta-policy's probability of selecting each policy head. Bottom row shows the number of treasures found per episode.

In Figure 19 we analyze the behavior of the meta-policy in two separate runs. We evaluate on Task 3, since we find that our method is able to surpass the best individual reward function. This task assigns specific goals to each agent. As such, one might expect that independent exploration would work most effectively in this setting. While independent exploration is effective (see Figure 10), we find that our method can outperform it. In both runs, we find that burrowing rewards are selected when the agents finally learn how to solve the task; however, we find that burrowing rewards are not necessarily successful when deployed on their own. This lack of success is likely due to the fact that these rewards cause the agents to pick a region and commit to exploring it for the duration of training. As such, the agents may pick the "wrong" region at first and never be able to recover. On the other hand, using our methods, the meta-policy can wait until the burrowing exploration regions align with the assigned rewards and then select the policies trained on these rewards. This usually ends up being more efficient than waiting for the agents to explore the whole map using independent rewards.

