# OpenReview forum: "Coordinated Exploration via Intrinsic Rewards for Multi-Agent Reinforcement Learning"
_ICLR.cc/2020/Conference — Reject_

### Official Review · AnonReviewer3 · 2019-10-22
**Official Blind Review #3**

**Rating:** 6

**Review:**

Contribution:

The paper proposes to use a set of handcrafted intrinsic rewards that depend on the novelty of an observation as perceived by the rest of the other agents. For each pair of reward and agent, they learn a policy and a value through actor critic method, and then a meta-policy choses at the beginning of each episode which intrinsic rewards to use, meaning that the policy used by the agents corresponds to the one that maximizes the reward chosen.



Review:


The major limitation of the paper in my opinion is the fact that the "coordination" that occurs here is only happening at training time, not at execution time. The agents eventually learn whatever trajectory they need to perform, and then proceed to do so without any interaction with the other agents. In a sense, they don't even learn to explore collaboratively. In other words, agents trained on task 1 in a given maze would not be able to solve task 2 on the same maze without essentially relearning everything from scratch.
The other corollary of the fact that each agent learns its own policy is that the number of agents is fixed at training time, preventing testing with a different number of agents, as sometimes done in the literature ([1] [2]).

Given this limitation the scope of the work basically reduces to the exploration of a fixed environment when the action space can be factored into different agents. This "multi-agent" formulation is presumably meant to break down the computational complexity of having a joint observation/action space. However, the experiments are conducted only with a very limited number of agents (only 2 in the non toy environment of vizdoom). This small scale doesn't, in my opinion, demonstrate the advantage of the decomposition of the MDP over say SOTA single-agent exploration methods applied to the cartesian product of all the agents action spaces (in vizdoom the paper considers only 3 actions, so with two agents it would amount to 9 actions, which is still very tractable). Once the trajectories of both agents are found, they can be distilled to each of them individually so that they only depend on the local observation.


Regarding the experiments on the Vizdoom environment, it appears that the traditional evaluation setup [3] doesn't involve providing the global position (x,y) to the agents as part of the observations (they must be inferred from the visual feed), contrary to the experimental setup presented in this paper.
In my opinion, this weakens the claim that the method "scales to more complex environments" since providing the position essentially makes the environment similar to a grid-world (arguably the visual feed isn't even needed to solve the task.


The use of a dynamic policy selection is somewhat interesting, but would benefit better investigation. Firstly, it is not clear to me if all the selection of the policy to use during training affects all the trajectories of the batch, or if different episodes of the batch may have a different policy.
Secondly, it seems that the setting is typically the one of a (non-stationary) bandit, since there is no state and the "reward" is the return obtained by the policy. Could you share the reason behind the choice of an actor-critic algorithm over classical bandit algorithms? One obvious advantage of the latter are provable regret bounds.
In all, the selection policy seems to be useful during training, since it sometimes yields better solutions than any of the individual reward schemes. It suggests that some form of curriculum over the rewards is occurring during training, but if this is really what is going on, then it's possible that the relevant literature about curriculum learning may offer more stable and principled solutions than an actor critic, for example population based training. This could potentially solve the issues observed in task 2.


[1] Relational Deep Reinforcement Learning, Zambaldi et al, https://arxiv.org/abs/1806.01830
[2] A Structured Prediction Approach for Generalization in Cooperative Multi-Agent Reinforcement Learning, Carion et al, https://arxiv.org/abs/1910.08809
[3] Curiosity-driven Exploration by Self-supervised Prediction, Pathak et al, ICML 2017


**Experience Assessment:**

I have published one or two papers in this area.

**Review Assessment: Checking Correctness Of Derivations And Theory:**

I assessed the sensibility of the derivations and theory.

**Review Assessment: Checking Correctness Of Experiments:**

I carefully checked the experiments.

**Review Assessment: Thoroughness In Paper Reading:**

I read the paper at least twice and used my best judgement in assessing the paper.

---

> ### Author Response · Authors · 2019-11-15
> **Thank You**
>
> Thank you for the detailed and insightful comments. Concerning the point of coordination only happening at training time, this is in fact by design. We are operating within the recently popularized regime of centralized training with decentralized execution [1,2,3], while the works referenced by the reviewer are concerned with learning a centralized controller. Within this regime, communication between agents is assumed to not be possible at testing time, so agents must learn to operate independently and coordination must be incorporated somehow during training. Furthermore, we are not concerned with learning policies that transfer between tasks. As is common in the intrinsic motivation literature, we simply want to explore the environment in order to enable learning in a single task with sparse rewards.
>
> 	The multi-agent formulation is not designed to break down the computational complexity of the problem, but rather our method is designed to be useful in multi-agent scenarios by incorporating inductive biases regarding spatial coordination. While the suggestion of learning with the joint action space and distilling to decentralized policies could potentially work for the 2 agent VizDoom case, such an approach would not work even in the 4 agent gridworld setting, where the joint action space contains 625 actions. An approach that would not work in this “toy” environment is not particularly useful to the multi-agent RL community. Furthermore, we have run experiments where we maintain our centralized training with decentralized execution paradigm but use intrinsic rewards where the agents are treated as one agent (this was also asked for by R1). Concretely, we use the inverse count of all agents being at their current positions, rather than considering agents independently. We refer the reviewer to the curves labelled “Centralized” in Appendix section A.5. We find that, while this approach is somewhat effective, it does not learn as quickly as our approach since it does not induce agents to coordinate their regions of exploration.
>
> 	In VizDoom we do not provide the agents’ locations as part of the observations. As described in the appendix, we provide the egocentric image view, along with an indicator of whether each reward has been collected. The global state, however, does include the agents’ locations and orientations, but this information is only available to the critic and not available to the policies during execution.
>
> 	To clarify the use of the policy selector: it only affects the selection of policies to use during rollouts. During training, all policies are trained simultaneously using off-policy methods. The point regarding non-stationary bandits is an interesting one. In fact, it was our first instinct to use such methods for the policy selector; however, we found that the convergence guarantees rely on assumptions on the nature of how the reward distribution evolves. Since we cannot characterize the evolution of rewards received by each policy type very well, we resorted to the policy gradient method described, which we find works well empirically. We have provided additional analysis of the nature of the policy selector and how it enables our method to surpass the performance of individual intrinsic reward methods (Appendix section A.6). We have not considered the idea of applying curriculum learning methods to our approach, though it is interesting and worth exploring.
>
> [1] Foerster, Jakob N., et al. "Counterfactual multi-agent policy gradients." Thirty-Second AAAI Conference on Artificial Intelligence. 2018.
> [2] Lowe, Ryan, et al. "Multi-agent actor-critic for mixed cooperative-competitive environments." Advances in Neural Information Processing Systems. 2017.
> [3] Rashid, Tabish, et al. "QMIX: monotonic value function factorisation for deep multi-agent reinforcement learning." arXiv preprint arXiv:1803.11485 (2018).

---

### Official Review · AnonReviewer1 · 2019-10-23
**Official Blind Review #2**

**Rating:** 3

**Review:**

Overall I like the approach in the paper. It proposes a nice 2 pronged method for exploiting exploration via intrinsic rewards for multi-agent systems. The parts that a bit lacking with the current version of the paper in this are the evaluation tasks are few and a bit simple and I think there needs to be more discussion on the "coverage" of the intrinsic reward types. Are the ones proposed motivated by the tasks in the paper or are they sufficient for tasks in general?  Last using a more recent novelty metric could allow the method to work on more interesting/complex tasks.

More detailed feedback:
- It would be good to include more learning curves in the main text for the paper.
- The fact that applying intrinsic motivation to multi-agent simulations seems like a natural idea would be to convert the problem to a "single" agent problem to compare against the "normal" application of intrinsic rewards. This might be another baseline to consider for comparison.
- It says that all agents share the same replay buffer. Does this also imply that every agent is performing the same task there are just many agents? This does not make the problem very multi-agent with different goals. Would it affect the algorithm significantly to work on an environment where the agents have various types of goals?
- As is noted in the text, this method appears to work well in the centralized training scheme that many have adopted recently. However, It makes me wonder if there is a way to employ these exploration schemes in a non-centralized training form. The ability to ask other agents in the world about there preferences and novelty of states appears to be a strong assumption, especially in a multi-agent robotics problem.
- While the authors note that the intrinsic rewards used in this work are not comprehensive it would be good to note how comprehensive they are. Are there a few that were left out on purpose. Do the authours believe this set is sufficient. This statement makes it seem like the authors just tried a few options and found one that worked. It would be good to expand on this discussion more.
- More detail for Figure 1 would be helpful to understand the overall network design. While that figure it helpful maybe it would be good to include a version that goes into detail for the 2 agent environment. Then a more compressed n agent version can also be shown.
- The paper describes a policy selector that is a type of high-level policy for HRL. This design seems rather unique in that this part of the policy can optimizing for which intrinsic reward to toggle based on the extrinsic rewards observed. I like it. It is noted that entropy is important for this design. Can this be analyzed in an empirical way? Is this true for most environments/tasks?
- Task 2 seems a bit contrived. Is there another instance of this type of task elsewhere in another paper? It would be better to use more standard tasks if they are available.
- Before section 6.1 the paper is discussing rewards the are received. It would be good to more explicit about where these rewards are coming from. I think it is meant that these rewards are the extrinsic rewards but it does not say.
- As noted just before section 6.1 it seems for the collection of tasks 1-3 it is already obvious what types of intrinsic rewards should be used. It would be good to include more tasks where this decision is less obvious.
- Why are there "black holes" in the environment? Also if an agent steps into a black hole they are crushed never to be seen again. What you describe sounds more like a wormhole where one end is non-stationary... Also, can the agents detect the presence of a black hole in some way?
- It appears the novel metric is count based. While this can work in practice it seems a rather simple metric. Is it possible to use something more like ICM or RND that was referenced in the paper? Especially for the VizDoom environment?
- In table 2 where are some of the numbers bold? It would be good to include this information in the caption for the table.
- I am not sure if the discussion on the behaviours the intrinsic reward functions result in are very surprising. Maybe there is a more interesting behaviour that results from the combination of two intrinsic rewards?


**Experience Assessment:**

I have published in this field for several years.

**Review Assessment: Checking Correctness Of Derivations And Theory:**

I carefully checked the derivations and theory.

**Review Assessment: Checking Correctness Of Experiments:**

I carefully checked the experiments.

**Review Assessment: Thoroughness In Paper Reading:**

I read the paper thoroughly.

---

> ### Author Response · Authors · 2019-11-15
> **Thank You**
>
> Thank you for the detailed and insightful comments.  We would like first to articulate more clearly the motivation behind the tasks we have used. When we consider the setting of multi-agent systems, we believe the notion of spatial coordination is of natural interest. In other words, should agents in such environments coordinate their spatial positioning while exploring. Our tasks are designed to test the extremes of the spectrum of spatial coordination (i.e. close together (task 2) vs. far apart (task 1)).
>
> When designing intrinsic reward functions, we focus on inducing spatial coordination in a general manner that does not use domain-specific knowledge. We argue that using these types of rewards for spatially coordinated exploration is generally useful for many problems in multi-agent systems.  The key contribution of our work is to provide examples which introduce a framework for expressing intrinsic reward functions that induce spatially coordinated exploration.
>
> Regarding the use of more recent novelty metrics: we *thoroughly* tested Random Network Distillation [2] on the ViZDoom tasks and concluded that RND is unsuitable for the domain. It is especially sensitive to the brightness of the input images, causing heavily visited rooms in the map with brighter walls leading to higher intrinsic rewards than unvisited dark colored rooms. Image preprocessing techniques (contrast limited adaptive histogram equalization) improved performance somewhat, but the exploration was still highly biased towards specific regions. We also implemented and tested the Intrinsic Curiosity Module [3] though we have not been able to replicate the results from the original paper and thus, cannot comment on its efficacy within our framework at this time.
>
> Below we will answer specific questions that were not answered above:
>
> -Comparing to single-agent intrinsic rewards
>
> This is an interesting point, and we have run the appropriate experiments to test its efficacy. We implement an intrinsic reward that considers the joint position of all agents (i.e. the inverse count of all agents being in their combined positions). Our initial hypothesis is that our intrinsic reward functions provide additional inductive biases which encourage spatial coordination and should outperform this standard centralized intrinsic reward function. Figures and a description of the implementation of centralized rewards are provided in the Appendix. We find that across all tasks with 2 agents our approach learns more efficiently than this baseline, confirming our hypothesis.
>
> -Agents sharing the same replay buffer
>
> We apologize for the confusion. Agents have multiple policy heads for each type of intrinsic reward, and these heads share experiences; however, agents do not share experiences with each other.
>
> -Is it possible to decentralize training?
>
> This is an interesting question. One can imagine an adaptation of our method where agents leave “markers” of how novel a region is to themselves such that other agents can detect these markers when they enter the region. In this way, the assumption of communicating the novelty of observations across agents can be removed and our approach can be adapted to a decentralized training regime. We leave this for future work.
>
> -Can the importance of entropy in the policy selector be analyzed empirically?
>
> We include experiments of our method run without the entropy bonus on the meta-policy across all tasks with 2 agents in the Appendix. We find that its removal consistently hurts performance across all settings.
>
> -Clarifying rewards described before section 6.1
>
> Yes, we are referring to the extrinsic rewards. The text has been modified to make this more clear.
>
> -Clarifying “black holes”
>
> These exist in order to make the exploration problem more challenging. Agents are able to detect the probability of any black hole in their immediate vicinity opening at the next time step, and must learn to navigate around them (i.e. wait for the probability of opening to be low).
>
> -Bolded numbers
>
> The bolded numbers are those where the best mean score falls within one standard deviation of their score distribution. We have added this description to the caption.
>
>
> [1] Taïga, Adrien Ali, et al. "Benchmarking bonus-based exploration methods on the arcade learning environment." arXiv preprint arXiv:1908.02388 (2019).
> [2] Burda, Yuri, et al. "Exploration by random network distillation." arXiv preprint arXiv:1810.12894 (2018).
> [3] Pathak, Deepak, et al. "Curiosity-driven Exploration by Self-supervised Prediction." International Conference on Machine Learning. 2017.

---

### Official Review · AnonReviewer2 · 2019-10-26
**Official Blind Review #2**

**Rating:** 3

**Review:**

Summary:
The paper proposes a method for coordinating the exploration efforts of agents in a multi-agent reinforcement learning setting. The approach has two main components: (i) learning different exploration policies using different "joint" intrinsic rewards; and (ii) learning a higher-level policy that selects one of the exploration policies to be executed at the beginning of each episode.

Each agent has its own novelty function which quantifies the novelty of observation seen by that agent. To coordinate exploration, these novelty functions are combined using aggregation functions to produce intrinsic reward for the agent. Each such aggregating function yields a different intrinsic reward. The authors propose several such aggregating functions as examples, however the method is applicable to other aggregating functions as well, as long as they can be computed off-policy.

During training, the higher level policy selects one of the exploration policies which is then executed for the entire episode. The episode data is used in two ways: (i) to train the higher-level policy using policy gradients for maximizing extrinsic rewards along with an entropy term; and (ii) to train each exploration policy using soft actor-critic on its own intrinsic reward function (and extrinsic reward) in an off-policy manner.

Experiments done on grid-world and VizDoom environment for three different tasks demonstrate that, on most tasks, the proposed approach performs at least as well as separately trained individual intrinsic rewards. Further ablation studies confirm that both the hierarchical setup and the "joint" intrinsic rewards are useful.


Questions to the Authors:

1. The second sentence in section 5 is not clear - "Furthermore, the type of reward ... sufficiently complex". The high-level policy selects an exploration strategy at the beginning of each episode and then sticks to it for the entire duration of the episode. Changing the exploration strategy over the course of training might be useful in cases when agent needs to switch to a different exploration strategy after reaching a particular bottleneck state. However, this would require the exploration strategy to be changed in the middle of an episode which is not supported. Could you give an example where the exploration strategy must be changed over time even if one only selects the strategy at the beginning of each episode? Also, why not select the exploration strategy after every fixed number of time steps within each episode (by making high-level policy a function of the current state)?

2. Analyzing the role of high-level policy and its evolution over time on different tasks would be a very nice addition to the paper. Qualitative experiments demonstrating that it provides a curriculum which helps the agents in surpassing the performance of individual intrinsic rewards would be helpful.

3. Should \Pi in (10) also depend on i?

Though paper is reasonably well written I find the contributions are very marginal. If authors can position the paper well with the existing literature and bring out the impact of the contributions it will be helpful.


**Experience Assessment:**

I have published one or two papers in this area.

**Review Assessment: Checking Correctness Of Derivations And Theory:**

N/A

**Review Assessment: Checking Correctness Of Experiments:**

I assessed the sensibility of the experiments.

**Review Assessment: Thoroughness In Paper Reading:**

I read the paper thoroughly.

---

> ### Author Response · Authors · 2019-11-15
> **Thank You**
>
> Thank you for the detailed and insightful comments. To address your first point, we can provide an illustrative example. Let us say that we are attempting to solve task 1, where agents must spread out and collect all rewards on the map. In this case it is possible that Leader-Follower rewards may be most successful at first since they all explore similar areas simultaneously, so if they happen to reach a region where a reward is located, there will be multiple agents there to have a chance at collecting it; however, this behavior is not optimal, as there are other rewards to collect around the map. At this point in training, burrowing rewards may become more optimal since at least one of the agents will continue exploring the same region to collect the reward, but the other agents will explore other regions, potentially collecting more rewards. We find that this type of situation occurs with reasonable frequency during training. Re-selecting the exploration strategy within each episode may be useful for tasks with multi-stage goals, though we don’t consider these types of tasks in this work.
>
> 	We have analyzed the role of the policy selector and have found some interesting behaviors. We refer the reviewer to the revised Appendix (section A.6) for a discussion on this analysis. Finally, the \Pi in equation 10 (i.e. the policy selector) does not depend on i, as all agents roll out their policies trained on the same intrinsic reward simultaneously. In other words, there is no mixing of different policy types.
>
> 	With respect to the position of our work within the literature, we believe we are the first work to address the problem of spatially coordinated exploration in deep multi-agent reinforcement learning. Many multi-agent tasks require some notion of spatial coordination for optimal performance (e.g. search-and-rescue), and our method induces such coordination in exploration with decentralized agents, enabling success in these types of tasks with sparse rewards. We show that naive applications of single-agent methods (Figure 3b independent and centralized intrinsic rewards) to multi-agent systems are relatively ineffective when compared to our approach.

---

### Decision · Program_Chairs · 2019-12-19

**Decision:**

Reject

**Comment:**

The authors present a method that utilizes intrinsic rewards to coordinate the exploration of agents in a multi-agent reinforcement learning setting.   The reviewers agreed that the proposed approach was relatively novel and an interesting research direction for multiagent RL.  However, the reviewers had substantial concerns about writing clarity, the significance of the contribution of the propose method, and the thoroughness of evaluation (particularly the number of agents used and limited baselines).  While the writing clarity and several technical points (including addition ablations) were addressed in the rebuttal, the reviewers still felt that the core contribution of the work was a bit too marginal.  Thus, I recommend this paper to be rejected at this time.